# Intracervical Foley Catheter Plus Intravaginal Misoprostol vs Intravaginal Misoprostol Alone for Cervical Ripening: A Meta-Analysis

**DOI:** 10.3390/ijerph17061825

**Published:** 2020-03-11

**Authors:** Howard Hao Lee, Ben-Shian Huang, Min Cheng, Chang-Ching Yeh, I-Chia Lin, Huann-Cheng Horng, Hsin-Yi Huang, Wen-Ling Lee, Peng-Hui Wang

**Affiliations:** 1Department of Obstetrics and Gynecology, Taipei Veterans General Hospital, Taipei 112, Taiwan; l.harvee@gmail.com (H.H.L.); benshianhuang@gmail.com (B.-S.H.); alchemist791025@gmail.com (M.C.); ccyeh39@gmail.com (C.-C.Y.); iclin6@vghtpe.gov.tw (I.-C.L.); hchorng@vghtpe.gov.tw (H.-C.H.); 2Department of Obstetrics and Gynecology, National Yang-Ming University, Taipei 112, Taiwan; 3Biostatics Task Force, Taipei Veterans General Hospital, Taipei 112, Taiwan; sweethsin509@gmail.com; 4Institute of Clinical Medicine, National Yang-Ming University, Taipei 112, Taiwan; 5Department of Medicine, Cheng-Hsin General Hospital, Taipei 112, Taiwan; 6Department of Nursing, Oriental Institute of Technology, New Taipei City 220, Taiwan; 7Department of Medical Research, China Medical University Hospital, Taichung 440, Taiwan; 8The Female Cancer Foundation, Taipei 104, Taiwan

**Keywords:** induction, intracervical Foley catheter, intravaginal misoprostol, labor, term pregnancy

## Abstract

Currently, there is no meta-analysis comparing intravaginal misoprostol plus intracervical Foley catheter versus intravaginal misoprostol alone for term pregnancy without identifying risk factors. Therefore, the purpose of this study is to conduct a systematic review and meta-analysis of randomized control trials (RCTs) comparing concurrent intravaginal misoprostol and intracervical Foley catheter versus intravaginal misoprostol alone for cervical ripening. We systematically searched Embase, Pubmed, and Cochrane Collaboration databases for randomized controlled trials (RCTs) comparing intracervical Foley catheter plus intravaginal misoprostol and intravaginal misoprostol alone using the search terms “Foley”, “misoprostol”, “cervical ripening”, and “induction” up to 29 January 2019. Data were extracted and analyzed by two independent reviewers including study characteristics, induction time, cesarean section (C/S), clinical suspicion of chorioamnionitis, uterine tachysystole, meconium stain, and neonatal intensive care unit (NICU) admissions. Data was pooled using random effects modeling and calculated with risk ratio (RR) and 95% confidence interval (CI). Pooled analysis from eight studies, including 1110 women, showed that labor induction using a combination of intracervical Foley catheter and intravaginal misoprostol decreased induction time by 2.71 h (95% CI −4.33 to −1.08, *p* = 0.001), as well as the risk of uterine tachysystole and meconium staining (RR 0.54, 95% CI 0.30–0.99 and RR 0.48, 95% CI 0.32–0.73, respectively) significantly compared to those using intravaginal misoprostol alone. However, there was no difference in C/S rate (RR 0.93, 95% CI 0.78–1.11) or clinical suspicion of chorioamnionitis rate (RR 1.22, CI 0.58–2.57) between the two groups. Labor induction with a combination of intracervical Foley catheter and intravaginal misoprostol may be a better choice based on advantages in shortening induction time and reducing the risk of uterine tachysystole and meconium staining compared to intravaginal misoprostol alone.

## 1. Introduction

Labor induction (induction of labor-IOL), or initiating labor before spontaneous onset of labor in a viable pregnancy, is often considered when the benefits of induction outweigh the risks of continued pregnancy or at the request of the pregnant women at term [1,2,3,4,5,6,7,8,9,10,11,12,13,14,15,16,17,18,19,20,21,22,23]. In modern obstetrics, it is an increasingly common practice and offers a better care for both fetus and mother. The World Health Organization (WHO) Global Survey on Maternal and Perinatal Health, which surveyed 24 countries, showed that 9.6% of deliveries involved labor induction and the induction rate in the United States has more than doubled in 2006 at 22.5% of all deliveries [11,24].

Cervical ripening is an important factor for a successful induction. Unripe cervix with a lower Bishop score is associated with an increased risk of induction failure, while a favorable cervix significantly predicts a timely delivery [25]. There are two main methods of cervical ripening. One is mechanical, including (1) the introduction of a catheter through the cervix into the extra-amniotic space with balloon insufflation; (2) introduction of laminaria tents, or their synthetic equivalent (Dilapan), into the cervical canal; (3) use of a catheter to inject fluid into the extra-amniotic space (EASI), and the other is pharmacological [1,7,8,14,15,26,27,28,29,30]. Intracervical Foley catheter is the most common mechanical method that was first described by Embrey and Mollison in 1967, where a Foley is inserted into the cervical canal and dilated just past the internal os with mild traction outward dilating the cervix directly, as well as indirectly stimulating prostaglandin (PG) and oxytocin secretion [31,32,33,34,35,36,37,38]. Pharmacological methods include many agents, such as PG (PGE2 or PGE1), progesterone receptor antagonists (mifepristone), oxytocin, and nitric oxide (NO) donors, but the most commonly used are prostaglandins and oxytocin. Misoprostol is a synthetic PGE1 analog and is used frequently, because of low cost and easy preservation (refrigeration is not needed), although it is not licensed for labor induction [6,10,19,39,40]. It achieves cervical softening through disintegration and dissolution of extracellular collagen [41]. There are multiple routes of administration for misoprostol, including oral, buccal, sublingual, vaginal, and rectal [41,42,43,44,45,46,47]. Compared to an oral or sublingual route, which creates a rapid spike in misoprostol plasma concentrations more beneficial in postpartum hemorrhage (PPH) management, intravaginal administration has a more gradual onset and greater bioavailability [45]. It has also been shown to be more effective in other obstetric treatments, such as abortion [43].

Many studies have compared the two methods, intravaginal misoprostol and intracervical Foley catheter, individually or when used concurrently [48,49,50,51,52,53,54,55,56,57,58,59]. It has been proposed that the two different mechanisms may have synergistic effects, as found by Al-Ibraheemi Z et al. and numerous other authors [48,49,50,52,53]. However, there have been other studies that showed a lack of such synergistic effect [54,55], while another study showed no significant benefits to the induction–delivery time but did reduce complications such as uterine hyper stimulation [51]. A meta-analysis in 2015 by Chen et al. showed that combined use of Foley catheter plus misoprostol is associated with a shorter mean time to delivery and reduced tachysystole, but increased clinical suspicion of chorioamnionitis when compared to misoprostol alone, but it included oral misoprostol [16]. Furthermore, Chen’s meta-analysis included studies with premature rupture of membrane, which affects induction times. Another meta-analysis by Eikelder et al. in 2016, which looked mainly at the safety profiles for both induction methods, showed that there was less uterine hyperstimulation in the combination group [15]. Moreover, a subgroup analysis with Foley catheter plus 25 μg vaginal misoprostol versus 25 μg vaginal misoprostol only also found less hyperstimulation in the combination group, although it did not reach statistical significance. However, time to delivery was not analyzed in this meta-analysis [15]. A 2019 Cochrane review was also performed encompassing several mechanical methods and pharmaceutical methods as well as extensive analysis various combinations including “any mechanical methods and low dose misoprostol versus low dose misoprostol alone” [1]. In this analysis, laminaria tent, balloon catheters extra-amniotic infusions were all grouped as one intervention. Currently, there is no meta-analysis specifically comparing intravaginal misoprostol plus intracervical Foley catheter versus intravaginal misoprostol alone for term pregnancy without identifying risk factors. Therefore, the purpose of this study is to conduct a systematic review and meta-analysis of randomized control trials (RCTs) comparing concurrent intravaginal misoprostol and intracervical Foley catheter versus intravaginal misoprostol alone for cervical ripening. This systematic review and meta-analysis was registered with PROSPERO, and PRISMA protocol was followed (ID:164559).

## 2. Materials and Methods

### 2.1. Data Sources and Searches

A literature search was conducted on PubMed, Cochrane Collaboration, and Embase databases up to January 29, 2019 for randomized controlled trials using the search terms “Foley”, “misoprostol”, “cervical ripening”, and “induction”. Reference lists from relevant articles, reviews, and meta-analyses were further screened and a cursory search on Google Scholar performed for potential articles.

### 2.2. Study Selection

Studies were included if they were of RCT design comparing combined intracervical Foley catheter and intravaginal misoprostol versus intravaginal misoprostol only for the purpose of cervical ripening and induction in viable singleton pregnancies. Studies that included ruptured membranes or intrauterine demise were not accepted, nor were studies that used other concurrent induction agents such as oxytocin, oral misoprostol or other PGs. Studies containing oxytocin used sequentially after last dose of intravaginal misoprostol or after the allotted time for intracervical Foley catheter were accepted. Additionally, studies with more than two intervention groups were not excluded as long as one of the groups included combined intravaginal misoprostol plus intracervical Foley catheter, and another intravaginal misoprostol only. Due to the nature of the interventions, the studies were not required to be blinded.

Two authors (H.H.L. and P.-H.W.) independently reviewed potentially eligible studies for inclusion. Disagreements were resolved by discussion until a consensus was reached.

### 2.3. Data Extraction

Data was extracted independently by two authors onto a structured excel sheet which included data on author, year, intravaginal misoprostol dosage, intracervical Foley balloon size and balloon volume, number of patients analyzed in each group, pregnancy duration, Bishop score, and outcomes. Differences in data between the two authors were reassessed and discussed until a consensus was reached. Attempts were made to contact authors of studies that lacked full text from conference posters or where clear definition of outcome and other relevant data was missing.

Quality assessment was performed using the Revised Cochrane risk-of-bias tool for randomized trials (RoB 2), 9 October 2018 version, and its modification in 2019 [60,61] by the two authors independently and differences resolved after discussion by consensus. The third author was brought in if no consensus could be reached. Five domains were assessed, including randomization process, deviations from intended interventions, missing outcome data, measurement of the outcome, and selection of the reported result. Questions pertaining to blinding in domain 2 and 4 were assessed as not applicable since the distinct nature of the interventions precluded blinding in the study design. Furthermore, for the last domain, selection of the reported results, pre-trial protocol had to be obtained for full assessment, which was not readily available for the majority of studies. Therefore, quality assessment without the last domain was also performed. Studies were categorized into low risk of bias, some concern, and high risk of bias. Studies were considered of low quality if any of the domains were assessed to be high risk for bias.

### 2.4. Selection of Outcomes

The primary outcome of this study was mean time from start of induction to delivery. Secondary outcomes included cesarean section rate, clinical suspicion of chorioamnionitis, uterine hyperstimulation, tachysystole with fetal heart rate changes, meconium stained liqueur, and admission to the neonatal intensive care unit (NICU). Many of the clinical outcomes were defined differently in each of the studies so data was collected according to the definition provided by each of the authors. For example, most studies defined chorioamnionitis in regard to clinical presentation rather than pathological. Therefore, chorioamnionitis in the context of this study is a clinical suspicion in line with the definitions used in the studies included in the meta-analysis.

### 2.5. Statistical Analysis

Statistical analyses were performed using Comprehensive Meta-Analysis Version 3.3.070 (Biostat inc. Englewood, NJ, USA, 20 November 2014). Overall mean difference and 95% confidence interval were estimated for continuous data such as time to delivery while summary risk ratios with 5% confidence interval were estimated for dichotomous data. All data was pooled using Mantel-Haenszel random-effects modeling. *p*-value was set at <0.05 for significance.

Then, I^2^ statistic and Cochrane Q test were used to assess heterogeneity between the studies using I^2^ > 50% and *p* < 0.10 respectively for significance. Sensitivity analysis was performed using different study characteristics as well as omitting one study at a time to evaluate the stability of results.

Publication bias was assessed using Funnel plots, Egger’s regression test, and Duval and Tweedie’s trim and fill test.

## 3. Results

### 3.1. Study Selection, Quality Assessment, and Study Characteristics

Of the 224 records found through database searches, 30 RCTs were identified and assessed for eligibility. Seventeen of the RCTs did not meet the inclusion criteria for the correct intervention and full text could not be obtained for 3 of the studies, leaving a total of 10 RCTs to be included in the meta-analysis [33,48,49,50,51,52,53,54,56,57] (Figure 1).

Among the 10 included studies, all were categorized as at least some concern for risk of bias due to the fifth domain requiring pre-trial protocol for assessment. Two of the studies were categorized as high risk. If risk is assessed without the fifth domain, 5 studies are considered low risk, 3 with some concerns, and 2 are categorized as high risk (Table 1). Two of the studies (Kashanian et al. 2006 [33] and Ashwini et al. 2018 [56]) had high risk of bias in almost all of the domains assessed, as well as overall assessment, so they were excluded from further analyses.

The characteristic of each trial is listed in detail in table 2. Pregnancy duration ranged from 24 weeks to >38 weeks and Bishop score ranged from <4 to <6. Most of the trials used 16 F urinary catheters except for one that used 18 F, another that used 24 F, and two that did not provide information on the size of catheter used. All of the trials used some form of traction for the intracervical Foley catheter except for one trial that did not specifically mention Foley traction. Misoprostol dosage ranged from 25μg every 4–6 h to 50μg every 6 h with max redosing from 4–8 times. Oxytocin was used sequentially and amniotomy use was up to the healthcare provider if mentioned at all (Table 2).

### 3.2. Time to Delivery

All 10 of the studies reported on induction to delivery time. However, Ugwu et al. reported the data in terms of number of patients per time interval so pooling into the meta-analysis was not possible [53]. Two of the papers reported results with median and range for induction to time of delivery so mean and standard deviation was estimated using a standard formula [62]. Levine et al. reported results without standard deviation so effects were estimated using mean, sample size, and *p*-value [49]. Two of the studies were excluded due to low quality of the studies.

Using random-effects estimation, the combination group was statistically significant for shorter mean time to delivery compared to intravaginal misoprostol alone by 2.705 h (*p* = 0.001) (Figure 2).

However, there was significant heterogeneity between the studies (*p* < 0.001, I^2^ = 94.256%) so several methods were used to identify the source of heterogeneity. First, each of the studies were systematically omitted and re-analyzed but no specific study accounted for the cause of the heterogeneity. Then subgroup analyses by study quality, Bishop score, gestational age, Foley size and misoprostol dose were conducted, all of which still had considerable heterogeneity. Finally, subgroup by sample size was performed, showing decreased heterogeneity when studies with over 200 subjects were put together and those with smaller sample sizes were grouped together (Figure 3 and Figure 4).

### 3.3. Cesarean Delivery

All eight of the studies analyzed included information on Cesarean delivery rate. There was no significant difference between the two groups using fixed effects model (Figure 5).

### 3.4. Uterine Tachysystole with and without Fetal Heart Rate Changes 

Eight of the nine studies provided data on uterine tachysystole with the definition >5 or 6 contractions within 10 min. However, three of the studies (Lanka et al. [51], Gilani et al. [57], and Ugwu et al. [53]) had no cases of uterine tachysystole in both groups so they were not included in the analysis. Using random effects model, there was less risk for uterine tachysystole with the combination group (Figure 6).

Four of the studies also provided information on fetal heart rate changes due to hyperstimulation. However, there were no events in either of the groups for Aduloju et al. [50] so it was not included in the analysis. Chung et al. [54] analyzed both uterine tachysystole and hyperstimulation where uterine tachysystole was defined as >6 contractions in 10 min for 2 consecutive 10 min periods and hyperstimulation was defined as tachysystole or hypertonus associated with new onset of prolonged or late decelerations. Carbone et al. defined tachysystole as >5 contractions in 10 min with decelerations requiring cessation of oxytocin infusion or terbutaline [52]. Lanka et al. defined it as >5 contractions in 10 min with fetal heart rate changes [51]. Upon further analysis of these three studies, there was a trend for higher risk of tachysystole with fetal heart rate changes, but it was not statistically significant, probably due to the small sample size (Figure 7).

### 3.5. Meconium Stain

Six of the studies had analysis for meconium stained amniotic fluid/liquor. Among them, only Al-Ibraheemi et al. stratified meconium stain into light, moderate, and thick [48]. For analysis, all three of the categories were included and pooled into the meta-analysis. The rest of the studies did not give in-depth definitions of meconium stained amniotic fluid/liquor nor provide further degrees of severity. For the 772 pregnant women analyzed, there was a significant increased risk for meconium stained amniotic fluid/liquor in the intravaginal misoprostol only group based on both fixed and random effects model (Figure 8).

### 3.6. Other Outcomes

There were no significant differences between the two groups for both risk of chorioamnionitis or neonatal intensive care unit (NICU) admission using fixed model. In regard to chorioamnionitis, four of the studies included relevant data [48,49,52,54]. One study by Lanka et al., which provided data on chorioamnionitis, was not included in the analysis since there were zero cases in both groups [51]. Chung et al. defined chorioamnionitis as body temperature >38.0 °C at any time during the induction process [54]. Levine et al. took into consideration maternal or fetal tachycardia and uterine tenderness as well as body temperature >38.0 °C [49]. The two remaining studies did not specify the definition of chorioamnionitis [48,52]. Using random effects model, there was no significant increased risk for chorioamnionitis between the two groups (Figure 9). As for NICU admission, criteria for admission were not specifically mentioned in any of the studies (Figure 10).

### 3.7. Publication Bias

Using Egger’s Test, there were no significant publication biases. Using Duval and Tweedie’s trim and fill analysis, adjusted risk ratios were not significantly different clinically to calculated risk ratio for any of the events analyzed (Table 3).

## 4. Discussion

This meta-analysis of 8 randomized controlled trials with 1110 women showed that combined use of Foley catheter and intravaginal misoprostol for induction of pregnancy leads to faster induction time with no increase in caesarean section rate as well as decreased risk for tachysystole and meconium when compared to intravaginal misoprostol alone. These findings are mostly in line with a previous 2015 meta-analysis, which also showed decreased induction time for combined Foley catheter and misoprostol use [16].

However, there are several strengths to our meta-analysis in contrast to the previous study. In this study, RCTs were assessed for risk of bias, and low quality studies were excluded from further analysis. The 2015 meta-analysis included one low quality RCT [33], which was also deemed of low quality under our analysis. Additionally, rupture of membrane may affect the speed for induction, so we have controlled for this factor by excluding all trials that include pregnant women with ruptured membranes in contrast to earlier meta-analysis. Furthermore, both oral as well as intravaginal misoprostol were used in the 2015 meta-analysis. To date, this is the only meta-analysis performed using a uniform administrative route, intravaginal, for misoprostol in combination with intracervical Foley catheter.

There are some limitations for this study. Not all randomized trials provided all the clinical outcomes analyzed in this meta-analysis decreasing the sample size in some of the outcomes measured. Some of the data reported were analyzed using different parameters making pooling difficult and further decreasing the number of papers included in each analysis. There are also variations in parity, Bishop score, pregnancy duration, and size of Foley catheter and dose regimen between each study. Furthermore, even though concurrent oxytocin use was excluded from this meta-analysis, difference in sequential oxytocin use may have affected the results. Other clinical circumstances, such as difficulty in Foley placement due to obesity or other unforeseen circumstances, are not accounted for. Moreover, the intravaginal route itself has a larger variation in uptake compared to other routes, which may be another source of variability [45].

The heterogeneity between the studies is also a concern. Sensitivity analysis according to patient characteristics, study quality, and systematic removal of one study did not account for the heterogeneity. However, size of the study did. When re-analyzed with subgroups according to size of the study with study size 200 as a divider, heterogeneity was greatly decreased. Quality of the studies was accounted for in our study, but pretrial protocols were difficult to obtain for the studies, which made it difficult to do a comprehensive risk of bias assessment. Moreover, the nature of the interventions precluded blinding from study design. Despite this, most of the studies included rated low for risk of bias.

Unlike Chen et al.’s analysis showing the increased risk of chorioamnionitis in women with intracervical Foley catheter [16], risk for chorioamnionitis was not increased with intracervical Foley catheter in our current study. This is perhaps due to the rupture of membrane being excluded from our study, since some studies have shown that rupture of membrane is a known risk factor for infection [63]. Despite this, another study from Battarbee et al. did not find an increased risk for any adverse maternal or neonatal outcome in early amniotomy after Foley balloon catheter removal for labor induction among nulliparous women [64]. An early 2011 meta-analysis by Fox et al. comparing intravaginal misoprostol only versus Foley catheter only has already shown no increased risk of chorioamnionitis [34]. The following 2015 meta-analysis from McMaster et al. evaluated 26 randomized trials including 5563 women to compare the risk of infectious morbidity between prostaglandin preparations alone and Foley catheter insertion alone and also failed to show an increased risk of chorioamnionitis in the Foley catheter insertion group (relative risk (RR) 0.96, 95% confidence interval (CI) 0.66–1.38) [17]. Of course, we still cannot totally exclude the baseline difference and/or bias among these studies, which might have contributed to the discrepancies of findings.

Tachysystole was significantly increased in the intravaginal misoprostol only group, which was also seen in the 2015 meta-analysis [17]. This may be due to the difference in total dosage of misoprostol between the two groups. Carbone et al. remarked that fewer doses of misoprostol were needed for the combination group [52] and in the study by Al-Ibraheemi et al., the combination group received statistically significant fewer misoprostol doses compared to the intravaginal misoprostol group [48]. However, it is difficult to come to a definite conclusion as apart from the above two studies, most studies did not provide the total dose of PGE_1_ for each group. Oxytocin usage may also affect the risk of tachysystole between the two groups. Of the RCTs analyzed, four studies used less oxytocin for the combination group whereas only one study, by Ugwu et al., used less oxytocin for misoprostol only group [53]. The rest of the studies did not mention total oxytocin needed until delivery. This provides a hint that combination use decreases oxytocin requirements, but again, more studies are needed for a more definitive conclusion.

As for tachysystole with fetal heart rate changes, only a trend favoring combined Foley and misoprostol use was seen but with only three studies included, further research with adequate power as well as more uniformed and detailed definition of fetal heart rate change is needed for a more conclusive result.

Interestingly, unlike any of the previous reviews and meta-analyses, risk for meconium stain was significantly increased in the misoprostol only group in this study. This may be explained by the increased tachysystole in the misoprostol-only group, stimulating bowel movement in the fetus. On the other hand, NICU admission was not significant in this meta-analysis, which is compatible with the previous 2015 meta-analysis [17].

In comparison to the 2019 Cochrane review, the findings of this study are mostly in line with the “Any mechanical method and low dose misoprostol versus low dose misoprostol alone” analysis. Whereas there was no significant decreased in vaginal deliveries achieved in 24 h there was a trend favouring combined mechanical and pharmaceutical cervical ripening methods. Adverse effects, such as uterine hyperstimulation without FHR changes, were also shown to decrease with combined use, as was with this study. There was also no increase in chorioamnionitis and endometritis rates in the Cochrane review [1]. However, it is important to keep in mind that all mechanical methods were included, such as laminaria tents, extra-amniotic infusion in addition to Foley catheters, and so the two studies cannot be directly compared.

Furthermore, it is important to note that none of the studies reported any uterine rupture so it was not possible to compare uterine rupture rate between the two methods, although this catastrophic situation should be always kept into mind [65,66]. In addition, the evidence based for cervical ripening for term induction, as well as caution of the using pharmacological agents or any mechanical methods for induction of labor should be always followed [67,68,69]. It should administer misoprostol cautiously using 25 μg every 4 h as recommended by the WHO and is also the dosage used by most of the included studies. The findings in this study are mostly in line with the 2019 Cochrane review, where Foley catheter in conjunction to vaginal misoprostol should be considered to decrease time to delivery as well as to decrease adverse effects associated with misoprostol, especially in countries where induction agents with better safety profiles, such as PGE2, are not available.

## 5. Conclusions

In pregnant women of viable gestation and no premature rupture of membranes, concurrent use of intracervical Foley catheter and intravaginal misoprostol shortens induction time and decreases risk of tachysystole compared to intravaginal misoprostol only without increasing risk for Cesarean section, chorioamnionitis, meconium stain, or NICU admission.

## Figures and Tables

**Figure 1 ijerph-17-01825-f001:**
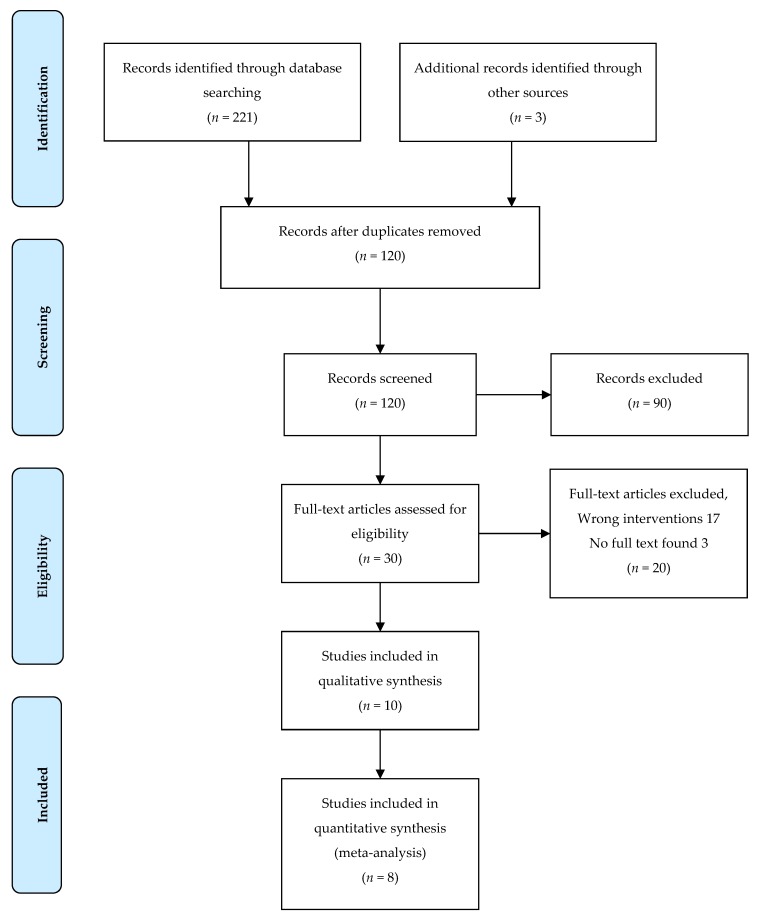
Flow chart of study selection.

**Figure 2 ijerph-17-01825-f002:**
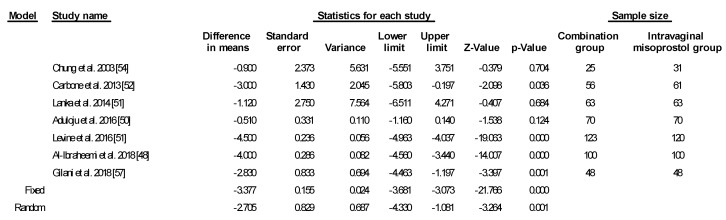
Comparison 1: Combination of intracervical Foley catheter and intravaginal misoprostol versus intravaginal misoprostol alone for cervical ripening, Outcome 1: Time to delivery. Heterogeneity: Tau^2^ = 3.500; χ^2^ = 104.455, df = 6 (*p* = 0.000); I^2^ = 94.256%. Test for overall effect (random): Z = −3.264, (*p* = 0.001).

**Figure 3 ijerph-17-01825-f003:**
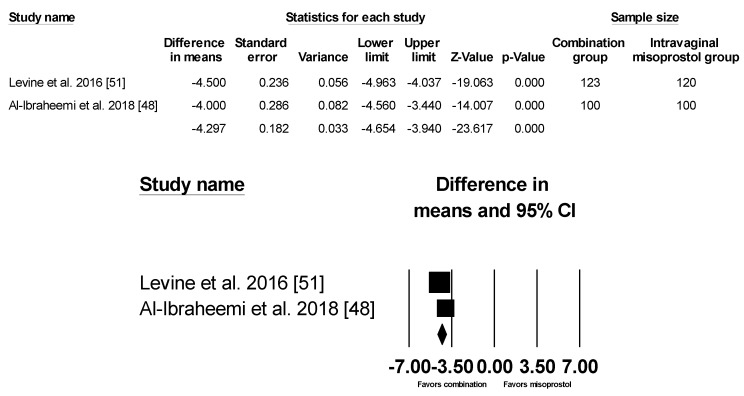
Comparison 2: Combination of intracervical Foley catheter and intravaginal misoprostol versus intravaginal misoprostol alone for cervical ripening, Outcome 2: Time to delivery in sample size > 200. Heterogeneity: Tau^2^ = 0.056; χ^2^ = 1.821 df = 1 (*p* = 0.177); I^2^ = 45.089%. Test for overall effect (fixed): Z = −23.167 (*p* = 0.000)

**Figure 4 ijerph-17-01825-f004:**
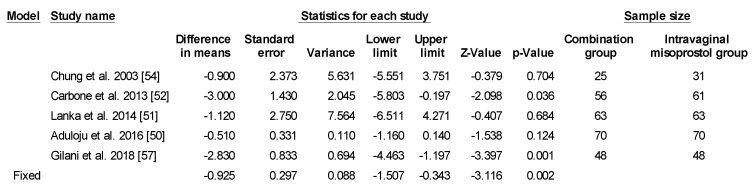
Comparison 3: Combination of intracervical Foley catheter and intravaginal misoprostol versus intravaginal misoprostol alone for cervical ripening, Outcome 3: Time to delivery in sample size < 200. Heterogeneity: Tau^2^ = 1.282; χ^2^ = 8.907 df = 4 (*p* = 0.063); I^2^ = 55.093%. Test for overall effect (random): Z = −2.215 (*p* = 0.027)

**Figure 5 ijerph-17-01825-f005:**
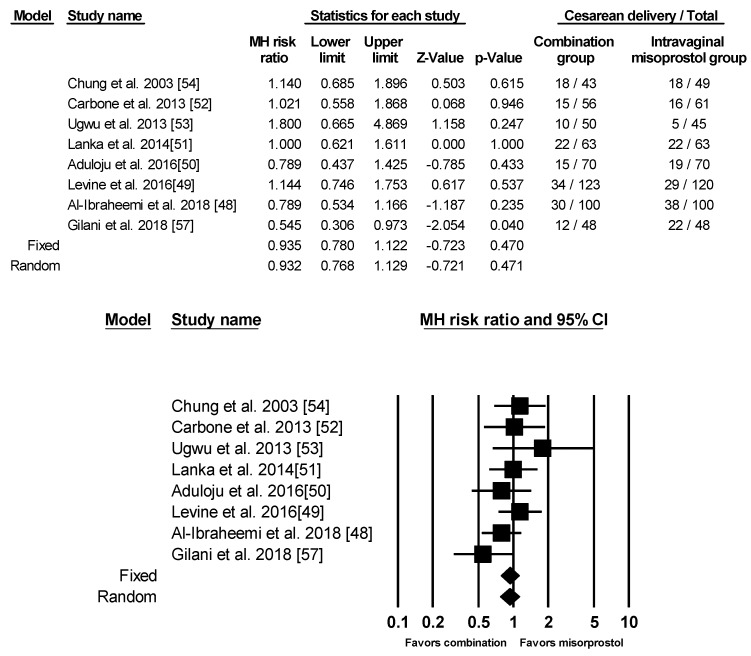
Comparison 4: Combination of intracervical Foley catheter and intravaginal misoprostol versus intravaginal misoprostol alone for cervical ripening, Outcome 4: Cesarean Section. Heterogeneity: Tau^2^ = 0.006; χ^2^ = 7.631, df = 7 (*p* = 0.366); I^2^ = 8.266%. Test for overall effect (fixed): Z = −0.723, (*p* = 0.470).

**Figure 6 ijerph-17-01825-f006:**
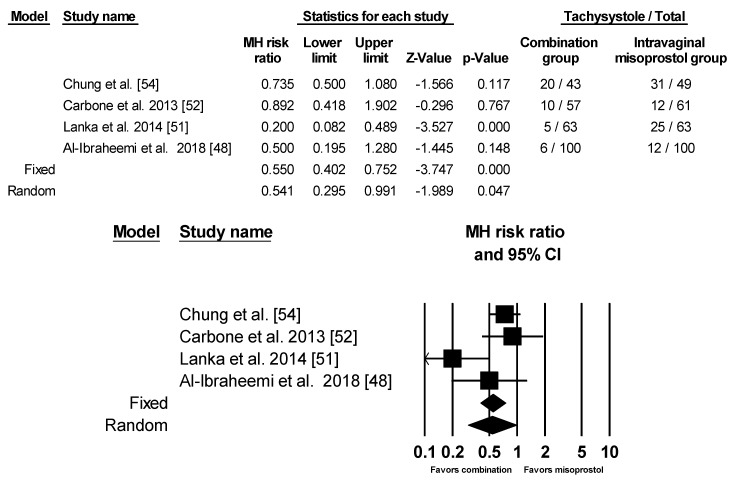
Comparison 5: Combination of intracervical Foley catheter and intravaginal misoprostol versus intravaginal misoprostol alone for cervical ripening, Outcome 5: Tachysystole. Heterogeneity: Tau^2^ = 0.242; χ^2^ = 8.705, df = 3 (*p* = 0.033); I^2^ = 65.536%. Test for overall effect (random): Z = −1.989, (*p* = 0.047)

**Figure 7 ijerph-17-01825-f007:**
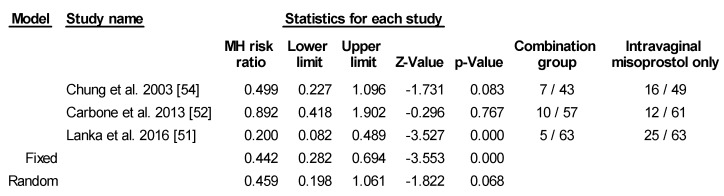
Comparison 6: Combination of intracervical Foley catheter and intravaginal misoprostol versus intravaginal misoprostol alone for cervical ripening, Outcome 6: Tachysystole with fetal heartbeat change. Heterogeneity: Tau^2^ = 0.377; χ^2^ = 6.406 df = 2 (*p* = 0.041); I^2^ = 68.778%. Test for overall effect (random): Z=−1.822 (*p* = 0.068).

**Figure 8 ijerph-17-01825-f008:**
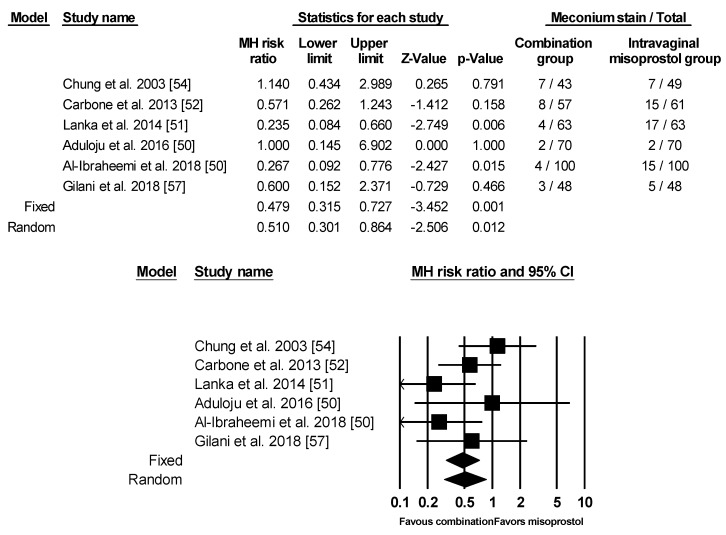
Comparison 7: Combination of intracervical Foley catheter and intravaginal misoprostol versus intravaginal misoprostol alone for cervical ripening, Outcome 7: Meconium stain. Heterogeneity: Tau^2^ = 0.119; χ^2^ = 6.942, df = 5 (*p* = 0.225); I^2^ = 27.972%. Test for overall effect (fixed): Z = −3.452 (*p* = 0.001)

**Figure 9 ijerph-17-01825-f009:**
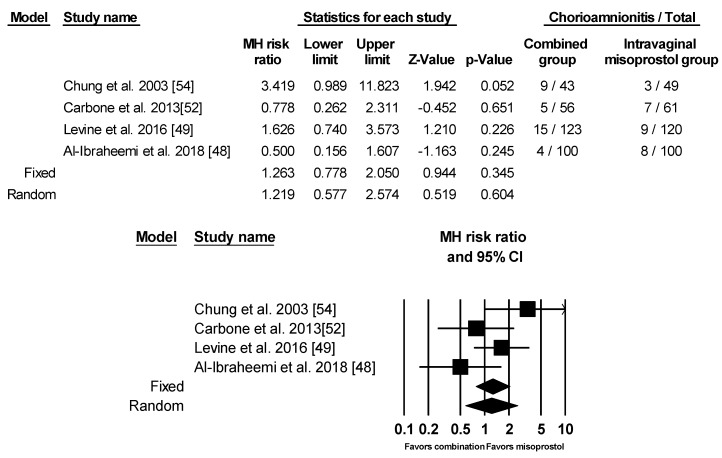
Comparison 8: Combination of intracervical Foley catheter and intravaginal misoprostol versus intravaginal misoprostol alone for cervical ripening, Outcome 8: Chorioamnionitis. Heterogeneity: Tau^2^ = 0.291; χ^2^ = 6.049, df = 3 (*p* = 0.109); I^2^ = 50.403%. Test for overall effect (random): Z = −0.519 (*p* = 0.604)

**Figure 10 ijerph-17-01825-f010:**
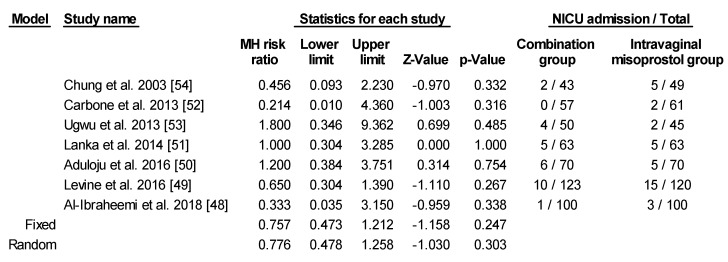
Comparison 9: Combination of intracervical Foley catheter and intravaginal misoprostol versus intravaginal misoprostol alone for cervical ripening, Outcome 9: neonatal intensive care unit (NICU) admission. Heterogeneity: Tau^2^ = 0.000; χ^2^ = 3.631 df = 6 (*p* = 0.726); I^2^ = 0.000%. Test for overall effect (fixed): Z = −1.158 (*p* = 0.247).

**Table 1 ijerph-17-01825-t001:** Study Risk of bias assessment.

Study	Rand Process *	Deviations	Miss *	Measure	Selection *	OB *	OB without Selection
Chung et al. 2003 [54]	Low	Low	Low	Low	Con.	Con.	Low
Kashanian et al. 2006 [33]	High	High	High	Low	Con.	High	High
Carbone et al. 2013 [52]	Low	Low	Con.	Low	Con.	Con.	Con.
Ugwu et al. 2013 [53]	Low	Low	Con.	Low	Con.	Con.	Con.
Lanka et al. 2014 [51]	Low	Low	Low	Low	Con.	Con.	Low
Aduloju et al. 2016 [50]	Low	Low	Low	Low	Con.	Con.	Low
Levine et al. 2016 [49]	Low	Low	Low	Low	Con.	Con.	Low
Al-Ibraheemi et al. 2018 [48]	Low	Low	Low	Low	Con.	Con.	Low
Ashwini et al. 2018 [56]	High	High	High	Low	Con.	High	High
Gilani et al. 2018 [57]	Con.	Low	Low	Low	Con.	Con.	Con.

* Pre-trial protocol could not be obtained so all studies were categorized as “some concerns: Con.”; Rand: randomization process; Deviations: deviations from the intended interventions; Miss: missing outcome data; Measure: Measurement of outcome; Selection: selection of the reported results; OB: overall bias.

**Table 2 ijerph-17-01825-t002:** Study characteristics.

Study	Inclusion Criteria		IA in F + M		IA in M	Oxytocin Used	AROM	No.	
	PD (gw)	BS	F size (mL)	M Dose and Frequency	Dose and Frequency			F + M	M
Chung et al. 2003 [54]	>28	<6	16 Fr. 30 mL, traction with tape to inner thigh, max 12 h	25 μg q3h, until AC, max 6 doses	25 μg q3h, until AC, max 6 doses	3 h later at the end of procedure	At cervical dilation > 3 cm	43	49
Kashanian et al. 2006 * [33]	>28	<5	16 Fr. traction with 500 mL NS	25 μg q3h, max 6 doses	25 μg q3h, max 6 doses	12 h after if absence of AC	NA	100	100
Carbone et al. 2013 [52]	>24	<6	Size (NA), 60 mL under gentle traction to inner thigh	25 μg q4h, until BS > 6, max 6 doses	25 μg q4h, until BS > 6, max 6 doses	4 h later at the end of procedure	Discretion	57	61
Ugwu et al. 2013 [53]	>37	<6	16 Fr. 30 mL, traction with tape to inner thigh, max 12 h, repeated once if BS < 5	25 μg q4h, until BS > 6, max 6 doses	25 μg q4h, until BS > 6, max 6 doses	4 h after last M later at the end of procedure or once reaching favorable BS	NA	50	45
Lanka et al. 2014 [51]	>28	<4	16 Fr. 30 mL, traction with tape to inner thigh, max 12 h	25 μg q4h, until BS > 6, max 8 doses	25 μg q4h, until BS > 6, max 8 doses	AM	AM before oxytocin was added	63	63
Aduloju et al. 2016 [50]	at term	<6	16 Fr. 30 mL, traction with tape to inner thigh, max 12 h, repeated once more if BS < 6	25 μg q6h, until BS > 6, max 4 doses	25 μg q6h, until BS >6, max 4 doses	6 h after last M later at the end of procedure or once reaching favorable BS	At cervical dilation > 4 cm	70	70
Levine et al. 2016 [49]	>37	<6	18 Fr. 60 mL, traction with tape to inner thigh, max 12 h	25 μg q3h, max 6 doses	25 μg q3h, max 6 doses	Started if additional cervical ripening was not indicated or at the end of procedure	Discretion (after cervix > 4 cm)	123	120
Al-Ibraheemi et al. 2018 [48]	>37	<6	Size (NA), 60 mL under gentle traction to inner thigh	25 μg q6h, until BS >6, max 8 doses	25 μg q6h, until BS >6, max 8 doses	Started if AC after last M	Discretion	100	100
Ashwini et al. 2018 * [56]	>28	<6	16 Fr. 50 mL under gentle traction to inner thigh	25 μg q4h, until cervix favorable, max 6 doses	25 μg q4h, until cervix favorable, max 6 doses	NA	NA	50	50
Gilani et al. 2018 [57]	>38	NA	16 Fr. 60 mL, traction with tape to inner thigh, max 12 h	50 μg q6h, until BS > 6, max 3 doses	50 μg q6h, until BS > 6, max 3 doses	NA	NA	48	48

IA: Induction agents; F: intracervical Foley balloon; M: intravaginal misoprostol; AROM: artificial rupture of membrane; No.: number of patients; PD: pregnancy duration; gw: gestational weeks; BS: Bishop Scores; q3h: every 3 h; q4h: every 4 h; q6h: every 6 h; max: maxima; mins: minutes; NA: no data available; AC: adequate contractions meaning >3 contractions/10 min: at the end of procedure: the last dose of misoprostol or Foley catheter expulsion or removal; AM: active management. * study was not included in final meta-analysis due to high risk of bias after quality assessment.

**Table 3 ijerph-17-01825-t003:** Publication bias.

Outcomes	Studies	Egger’s Test P Value	Trim and Fill, No. of Missing Studies	Calculated RR	Adjusted RR
Time to delivery	7	0.558	4	Random −2.705 (−4.330 to −1.081)	−4.136 (−5.845 to −2.428)
Sample size < 200	5	0.302	0	N/A	N/A
Cesarean Section	8	0.563	1	Fixed 0.930 (0.775 to 1.112)	0.910 (0.775 to 1.117)
Chorioamnionitis	4	0.825	0	NA	NA
Uterine tachysystole	4	0.414	0	NA	N/A
w/FHR change	3	0.124	0	NA	NA
Meconium Stain	6	0.881	1	Fixed 0.512 (0.332 to 0.789)	0.418 (0.282 to 0.621)
NICU	7	0.567	0	NA	NA

No.: number of patients; RR: risk ratio; NIC: neonatal intensive care unit’ NA: no data available; FHR: fetal heart rate.

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
