# Peer review of "Intracervical Foley Catheter Plus Intravaginal Misoprostol vs Intravaginal Misoprostol Alone for Cervical Ripening: A Meta-Analysis"

_ijerph, 2020, doi:10.3390/ijerph17061825_

Round 1

Reviewer 1 Report

Topic is novel.

English language is adequate.

Sample is large and this gives strenght to the study.

Material and Methods are well structured.

Results are well presented.

Discussion needs minor revision regarding better compare with other studies, as well as what will be the implication of this study to future clinical practice.

Author Response

Reviewer 1

Open Review

English language and style

( ) Extensive editing of English language and style required 
( ) Moderate English changes required 
(x) English language and style are fine/minor spell check required 
( ) I don't feel qualified to judge about the English language and style 

Yes

Can be improved

Must be improved

Not applicable

Does the introduction provide sufficient background and include all relevant references?

(x)

( )

( )

( )

Is the research design appropriate?

(x)

( )

( )

( )

Are the methods adequately described?

(x)

( )

( )

( )

Are the results clearly presented?

(x)

( )

( )

( )

Are the conclusions supported by the results?

(x)

( )

( )

( )

Comments and Suggestions for Authors

Topic is novel.

English language is adequate.

Sample is large and this gives strength to the study.

Material and Methods are well structured.

Results are well presented.

Discussion needs minor revision regarding better compare with other studies, as well as what will be the implication of this study to future clinical practice.

Response:

Dear reviewer, thank you for your constructive feedback.

We have already compared the results to a couple of more related studies, namely, the meta-analysis by Chen in 2015 (reference 16) and a 2011 meta-analysis by Fox et al. (reference 34). However, we recognize that we did not include the largest and most comprehensive and recent Cochrane review in the discussion. We have added the following to the discussion in order to address this: “ In comparison to the 2019 Cochrane review, the findings of this study are mostly in line with the “Any mechanical method and low dose misoprostol versus low dose misoprostol alone” analysis. Whereas there was no significant decreased in vaginal deliveries achieved in 24 hours there was a trend favouring combined mechanical and pharmaceutical cervical ripening methods. Adverse effects, such as uterine hyperstimulation without FHR changes were also shown to decrease with combined use as was with this study. There was also no increase in chorioamnionitis and endometritis rates in the Cochrane review [1]. However, it is important to keep in mind that all mechanical methods were included, such as laminaria tents, extra-amniotic infusion in addition to foley catheters and so the two studies cannot be directly compared.”

Furthermore, it is important to note that none of the studies reported any uterine rupture so it was not possible to compare uterine rupture rate between the two methods, although this catastrophic situation should be always kept into mind [66,67]. In addition, the evidence based for cervical ripening for term induction, as well as caution of the using pharmacological agents or any mechanical methods for induction of labor should be always followed [68-70] It should administer misoprostol cautiously using 25μg every 4 hours as recommended by the WHO and is also the dosage used by most of the included studies. The findings in this study are mostly in line with the 2019 Cochrane review where Foley catheter in conjunction to vaginal misoprostol should be considered to decrease time to delivery as well as to decrease adverse effects associated with misoprostol especially in countries where induction agents with better safety profiles such as PGE2 are not available.

Reviewer 2 Report

The manuscript represents a meta-analysis concentrating on one of the most popular obstetric procedures, i.e. cervical ripening. The authors try to compare two methods: intravaginal misoprostol vs. intravaginal misoprostol with intracervical Foley catheter. 

The misoprostol use for cervical ripening is contraindicated in the case of viable pregnancies in many countries. Taking into account the above-mentioned statement and the methodology of the reviewed manuscript (i.e. the study selection - viable singleton pregnancies, line 104), it seems that the interest to the readers could be limited.

Table 2 causes some confusion regarding the studies that were included in the meta-analysis. There are 10 studies with 1410 subjects, whereas the authors informed about eight studies with 1110 subjects (line 355).

Please correct the description of figures (for example line 270 and 314).

The statement that "Unripe cervix with a higher Bishop score is associated with an increased risk of induction failure" (lines 53-54) is controversial. Please explain.

Author Response

Reviewer 2

Open Review

English language and style

( ) Extensive editing of English language and style required 
( ) Moderate English changes required 
(x) English language and style are fine/minor spell check required 
( ) I don't feel qualified to judge about the English language and style 

Comments and Suggestions for Authors

The manuscript represents a meta-analysis concentrating on one of the most popular obstetric procedures, i.e. cervical ripening. The authors try to compare two methods: intravaginal misoprostol vs. intravaginal misoprostol with intracervical Foley catheter. 

The misoprostol use for cervical ripening is contraindicated in the case of viable pregnancies in many countries. Taking into account the above-mentioned statement and the methodology of the reviewed manuscript (i.e. the study selection - viable singleton pregnancies, line 104), it seems that the interest to the readers could be limited.

Table 2 causes some confusion regarding the studies that were included in the meta-analysis. There are 10 studies with 1410 subjects, whereas the authors informed about eight studies with 1110 subjects (Line 355).

Please correct the description of figures (for example, Line 270 and 314).

The statement that "Unripe cervix with a higher Bishop score is associated with an increased risk of induction failure" (lines 53-54) is controversial. Please explain.

Response:

Dear reviewer, thank you for your valuable input. It is true that misoprostol is relatively contraindicated in many countries. However, it is still the recommended method by the WHO. We feel that misoprostol as a cervical ripening agent in viable pregnancies is still a worthwhile topic for research especially for countries where medical resources are scarcer or where cost can be a burden.

As for table 2, we have clarified the two studies that were included in the qualitative assessment but not the final analysis due to high risk of bias by adding asterisks (*) to the two studies (Kashanian, 2006; and Ashwini, 2018) and added the following subtext: * study was not included in final meta-analysis due to high risk of bias after quality assessment.

Description of figures has been corrected.

In regards to the statement "Unripe cervix with a higher Bishop score is associated with an increased risk of induction failure", we have corrected it to "Unripe cervix with a lower Bishop score is associated with an increased risk of induction failure". (Line 53)

Finally, much more detailed information was provided in the discussion section to emphasize the safety of using any pharmacologic agents or mechanical tools for induction. Please read: 

In comparison to the 2019 Cochrane review, the findings of this study are mostly in line with the “Any mechanical method and low dose misoprostol versus low dose misoprostol alone” analysis. Whereas there was no significant decreased in vaginal deliveries achieved in 24 hours there was a trend favouring combined mechanical and pharmaceutical cervical ripening methods. Adverse effects, such as uterine hyperstimulation without FHR changes were also shown to decrease with combined use as was with this study. There was also no increase in chorioamnionitis and endometritis rates in the Cochrane review[1]. However, it is important to keep in mind that all mechanical methods were included, such as laminaria tents, extra-amniotic infusion in addition to Foley catheters and so the two studies cannot be directly compared.

Furthermore, it is important to note that none of the studies reported any uterine rupture so it was not possible to compare uterine rupture rate between the two methods, although this catastrophic situation should be always kept into mind [66,67]. In addition, the evidence based for cervical ripening for term induction, as well as caution of the using pharmacological agents or any mechanical methods for induction of labor should be always followed [68-70] It should administer misoprostol cautiously using 25μg every 4 hours as recommended by the WHO and is also the dosage used by most of the included studies. The findings in this study are mostly in line with the 2019 Cochrane review where Foley catheter in conjunction to vaginal misoprostol should be considered to decrease time to delivery as well as to decrease adverse effects associated with misoprostol especially in countries where induction agents with better safety profiles such as PGE2 are not available.

Reviewer 3 Report

This is a systematic review of the effectiveness of intracervical balloon plus  misoprostol v misoprostol alone. It suggests that there are advantages to the combination.  

It is well conducted. It was registered on PROSPERO.  I've checked against the Cochrane review of mechanical methods  https://www.cochranelibrary.com/cdsr/doi/10.1002/14651858.CD001233.pub3/epdf/full

and this particular comparison does not seem to have been done.  Since the control arm of misoprostol alone is the current recommendation and widely used this is potentially of clinical importance. Note the Cochrane review is reference 1 in the paper. But the authors do not explain specifically what is original about their review as compared with the Cochrane one. They do compare with their reference 15 which is a similar review by the same authors. 

I think they should bring the Cochrane review into their discussion. And say why the present review differs from both Ref 1 and ref 15.  

They state the review was registered with PROSPERO but a search on PROSPERO for the number they gave 164559 did not reveal it.  Can they give more details e.g. the url so readers can access it easily. 

Jim Thornton

Nottingham. UK  26 feb 2020

Author Response

Reviewer 3

Open Review

English language and style

( ) Extensive editing of English language and style required 
( ) Moderate English changes required 
(x) English language and style are fine/minor spell check required 
( ) I don't feel qualified to judge about the English language and style 

Yes

Can be improved

Must be improved

Not applicable

Does the introduction provide sufficient background and include all relevant references?

(x)

( )

( )

( )

Is the research design appropriate?

(x)

( )

( )

( )

Are the methods adequately described?

(x)

( )

( )

( )

Are the results clearly presented?

(x)

( )

( )

( )

Are the conclusions supported by the results?

(x)

( )

( )

( )

Comments and Suggestions for Authors

This is a systematic review of the effectiveness of intracervical balloon plus misoprostol v misoprostol alone. It suggests that there are advantages to the combination.  

It is well conducted. It was registered on PROSPERO.  I've checked against the Cochrane review of mechanical methods  https://www.cochranelibrary.com/cdsr/doi/10.1002/14651858.CD001233.pub3/epdf/full

and this particular comparison does not seem to have been done.  Since the control arm of misoprostol alone is the current recommendation and widely used this is potentially of clinical importance. Note the Cochrane review is reference 1 in the paper. But the authors do not explain specifically what is original about their review as compared with the Cochrane one. They do compare with their reference 15, which is a similar review by the same authors. 

I think they should bring the Cochrane review into their discussion. And say why the present review differs from both Ref 1 and ref 15.  

They state the review was registered with PROSPERO but a search on PROSPERO for the number they gave 164559 did not reveal it.  Can they give more details e.g. the url so readers can access it easily. 

Jim Thornton

Nottingham. UK  26 feb 2020

Response:

Dear Dr. Thornton, thank you for your review and valuable comments.

In regards to adding the Cochrane review into our manuscript as recommended, we have added the following to the introduction.  “A 2019 Cochrane review was also performed encompassing several mechanical methods and pharmaceutical methods as well as extensive analysis various combinations including “any mechanical methods and low dose misoprostol versus low dose misoprostol alone” [1]. In this analysis, laminaria tent, balloon catheters extra-amniotic infusions were all grouped as one intervention. Currently, there is no meta-analysis specifically comparing intravaginal misoprostol plus intracervical Foley catheter versus intravaginal misoprostol alone…”

We have also expanded on the discussion to include the following: “In comparison to the 2019 Cochrane review, the findings of this study are mostly in line with the “Any mechanical method and low dose misoprostol versus low dose misoprostol alone” analysis. Whereas there was no significant decreased in vaginal deliveries achieved in 24 hours there was a trend favouring combined mechanical and pharmaceutical cervical ripening methods. Adverse effects, such as uterine hyperstimulation without FHR changes were also shown to decrease with combined use as was with this study. There was also no increase in chorioamnionitis and endometritis rates in the Cochrane review [1]. However, it is important to keep in mind that all mechanical methods were included, such as laminaria tents, extra-amniotic infusion in addition to Foley catheters and so the two studies cannot be directly compared. “

Furthermore, it is important to note that none of the studies reported any uterine rupture so it was not possible to compare uterine rupture rate between the two methods, although this catastrophic situation should be always kept into mind [66,67]. In addition, the evidence based for cervical ripening for term induction, as well as caution of the using pharmacological agents or any mechanical methods for induction of labor should be always followed [68-70] It should administer misoprostol cautiously using 25μg every 4 hours as recommended by the WHO and is also the dosage used by most of the included studies. The findings in this study are mostly in line with the 2019 Cochrane review where Foley catheter in conjunction to vaginal misoprostol should be considered to decrease time to delivery as well as to decrease adverse effects associated with misoprostol especially in countries where induction agents with better safety profiles such as PGE2 are not available.

In regards to PROSPERO, we have registered it on 7 January 2020(https://www.crd.york.ac.uk/PROSPERO/record_email.php), and it is still undergoing reviewing process. Please see the detailed information shown below.

PROSPERO

International prospective register of systematic reviews

Intracervical Foley Catheter Plus Intravaginal Misoprostol vs Intravaginal Misoprostol Alone for Cervical Ripening: A Meta-analysis

From

To

Date

Subject

CRD-REGISTER

"[email protected]"

Tue, 7 Jan 2020 04:06:52 +0000

PROSPERO acknowledgement of receipt [164559]
